# Multimodal Language Learning for Object Retrieval in Low Data Regimes in the Face of Missing Modalities

**Kasra Darvish**                                                                    *kasradarvish@umbc.edu*
*University of Maryland, Baltimore County*

**Edward Raff**                                                                      *Raff_Edward@bah.com*
*University of Maryland, Baltimore County*
*Booz Allen Hamilton*

**Francis Ferraro**                                                                  *ferraro@umbc.edu*
*University of Maryland, Baltimore County*

**Cynthia Matuszek**                                                                 *cmat@umbc.edu*
*University of Maryland, Baltimore County*

**Reviewed on OpenReview:** *https://openreview.net/forum?id=cXa6Xdm0v7*

## Abstract

Our study is motivated by robotics, where when dealing with robots or other physical systems, we often need to balance competing concerns of relying on complex, multimodal data coming from a variety of sensors with a general lack of large representative datasets. Despite the complexity of modern robotic platforms and the need for multimodal interaction, there has been little research on integrating more than two modalities in a low data regime with the real-world constraint that sensors fail due to obstructions or adverse conditions. In this work, we consider a case in which natural language is used as a retrieval query against objects, represented across multiple modalities, in a physical environment. We introduce *extended multimodal alignment (EMMA)*, a method that learns to select the appropriate object while jointly refining modality-specific embeddings through a geometric (distance-based) loss. In contrast to prior work, our approach is able to incorporate an arbitrary number of views (modalities) of a particular piece of data. We demonstrate the efficacy of our model on a grounded language object retrieval scenario. We show that our model outperforms state-of-the-art baselines when little training data is available. Our code is available at https://github.com/kasraprime/EMMA.

## 1 Introduction

Humans use multiple modalities to understand and interact with the world around them. Although physical agents such as robots typically have access to multiple sensory inputs, most approaches use at most two senses (e.g., depth images along with standard RGB images) with single labels. Simultaneously using additional modalities and labels is an underexplored area. The benefits of integrating multiple modalities are twofold: first, complementary information from different modalities can help with understanding the world, and second, additional modalities can help when one or more sources of data become unavailable. Hardware can become damaged or defective, sensors can get blocked or obstructed, and various adverse conditions can remove a modality from use. However, designing an approach that integrates an arbitrary number of modalities while being robust to the loss of one or more modalities at test time is an important, yet non-trivial, problem. Meanwhile, the ability to learn from small amounts of training data has a significant impact on the adaptability and usefulness of robots. When they are deployed, they do not necessarily have access to large datasets to fine-tune themselves, particularly when they are attempting to interact with people. Therefore,

a guided learning method that makes use of signals in a low-data regime is crucial. Grounded language understanding, in which natural language is used as a query against objects in a physical environment, allows an intuitive mechanism by which users can instruct agents to engage in tasks. Visuolinguistic approaches to such object inference tasks typically involve training on large pools of image/text pairs and then using language to subselect elements of the sensed environment (Hong et al., 2021; Zhuge et al., 2021).

In order to fully leverage small amounts of multimodal training data, we propose a generalized distance-based loss function that can be extended to learn retrieval models that incorporate an arbitrary number of modalities, while being robust to missing information. We consider the domain of grounded language-based object retrieval (Hu et al., 2016; Nguyen et al., 2021), in which objects in an environment must be identified based on linguistic instructions. Previous approaches have explored combinations of sensors with labels provided by language (Richards et al., 2020). However, much of the existing work has not previously included an arbitrary number of modalities. The ultimate aim of this work is to take arbitrary input modalities about novel objects, including both spoken and written language, and build a model that allows a system to correctly retrieve that object given a subset of those input modalities.

In this work, we propose our new method called *extended multimodal alignment (EMMA)*, a generalized geometric method combined with a cross-entropy-based supervised contrastive loss function (Khosla et al., 2020). Our method can be used to learn retrieval models that incorporate an arbitrary number of views of a particular piece of data. We demonstrate steps towards this goal with a mechanism that learns quickly from a variety of input signals and is robust to the ablation of inputs. Our contributions are as follows:

1. We propose a new approach to multimodal learning that combines geometric and cross-entropy methods. Our method is designed to be applicable to additional modalities as they become available.
2. Our method performs better than state-of-the-art contrastive learning (Chen et al., 2020) and supervised contrastive learning (Khosla et al., 2020) especially when limited training data is available.
3. During training, our method converges about 2.3 times faster than the individual baselines.
4. We demonstrate robustness in the face of missing input signals when one or more modalities are ablated at test time, and propose a straightforward averaging method to quantify performance for multimodal object retrieval scenarios.
5. We demonstrate the separate utility of speech and text as sources of label information by treating them as sensory input modalities, rather than explicitly as labels.

## 2 Related Work

Grounded language learning (Roy & Pentland, 2002; Mooney, 2008; Matuszek et al., 2012) is intrinsically multimodal since its task is to learn the associations among at least two modalities of language and visual inputs. Prior work has focused on language and visual modalities, using a combination of language and vision to perform visual retrieval (Vo et al., 2019). While recent works have focused on grounding with respect to a knowledge set (Zheng & Zhou, 2019; Meng et al., 2020), our work focuses on robust multimodal learning to perform grounding over an arbitrary number of input modalities.

The growing number of datasets that contain modalities beyond images and text demonstrates the importance and the challenge of multimodal learning. Bagher Zadeh et al. (2018) introduce a large dataset of multimodal opinion sentiment and emotion intensity (CMU-MOSEI) for sentiment analysis and emotion recognition. Kébé et al. (2021) present a multimodal dataset of household objects containing RGB images, depth images, written language, and spoken language, which has been used to support learning grounded language directly from speech given small datasets (Kébé et al., 2022). Baltrušaitis et al. (2019) propose a taxonomy of multimodal machine learning by introducing five technical challenges including *alignment*—the process of identifying the direct relationship between elements across modalities—which is our focus here. Liang et al. (2022) also provide a thorough survey on multimodal learning challenges and approaches.

To the best of our knowledge, there has been limited work on incorporating more than three modalities into a general learning approach. Veit et al. (2018) train a joint model of images, hashtags, and users to perform image retrieval. This work differs from ours in that it becomes computationally complex as more and higher-dimensional modalities are added. Lei et al. (2020) use inputs from three modalities to perform image

retrieval given a sketch using a co-attention mechanism. Mittal et al. (2020) use Canonical Correlational Analysis (CCA) to differentiate between ineffective and effective modalities for the emotion recognition task from multiple input modalities including face, speech, and text. Abouelenien et al. (2014) use language, physiological response, and thermal sensing to detect deceit. However, recent work in the literature has given more attention to this problem (Liang et al., 2023; Reed et al., 2022). Liang et al. (2023) provide two new metrics to measure how similar two modalities are and how different pairs of similar modalities interact. The authors propose HighMMT, a multimodal transformer-based approach that can incorporate up to 10 modalities by fusing modalities that include unique information. Reed et al. (2022) proposed GATO, a generalized multimodal model that can perform multiple different tasks using the same network.

Variational autoencoders or VAEs (Kingma & Welling, 2013) are generative latent variable models that have had great success in generating images. Originally designed for unimodal tasks, later work has extended the technology to work in multimodal scenarios (Wu & Goodman, 2018; Shi et al., 2019; Palumbo et al., 2022). Transformer (Vaswani et al., 2017) is an attention-based neural network architecture consisting of an encoder and a decoder, originally proposed as a unimodal architecture for sequence modeling. Different varieties of transformer models gained attention after they proved to be successful in many different tasks (Lu et al., 2019; Raffel et al., 2020; Brown et al., 2020; Dosovitskiy et al., 2021; Lewis et al., 2020). It makes sense to use them for multimodal learning. Recent works have used multimodal transformers for a variety of tasks and modalities (*inter alia,* (Lu et al., 2019; Li et al., 2019; Likhosherstov et al., 2023; Zhang et al., 2023)).

Contrastive loss is mostly used in self-supervised learning (Bui et al., 2021; Alayrac et al., 2020; Chen et al., 2020), with some exceptions such as Khosla et al. (2020). However, a number of different approaches have been proposed for linguistic-based contrastive learning. Radford et al. (2021) introduced CLIP, which is designed to align two modalities of text and image using contrastive loss similar to SimCLR (Chen et al., 2020). Alayrac et al. (2020) use a self-supervised contrastive learning method to learn and combine representations from images, audio, and language. The loss function they define differs from ours in that they do not handle arbitrarily many modalities and do not focus on robustness in the face of modality drop-outs. Nguyen et al. (2020) take a similar approach and perform pairwise cosine similarity to align images and natural language text descriptions to perform object retrieval tasks. Nguyen et al. (2021) take a cross-modal manifold alignment approach to grounded language learning using triplet loss (Chechik et al., 2010) and perform object retrieval by connecting text descriptions of objects to their corresponding RGB-D images.

In contrastive learning, there are two broad families of approaches in the literature. One family is based on cross-entropy losses (Khosla et al., 2020; Chen et al., 2020), and the other is purely *geometric* (Poklukar et al., 2022; Carvalho et al., 2018; Salvador et al., 2017; Nguyen et al., 2021). In this paper we marry the two families and show that combining a geometric approach with cross-entropy based supervised contrastive loss is superior. The geometric approach is based on the idea of similarity (or distance) of elements of training data (Carvalho et al., 2018; Nguyen et al., 2021; Salvador et al., 2017). Our geometric loss function formulation is similar to lifted structured loss (Song et al., 2016) and quadruplet loss (Chen et al., 2017; Tursun et al., 2021), but we apply our idea to four modalities (and could incorporate more) and do not compute the distance among all items in a batch. Poklukar et al. (2022) take a geometric approach which is similar to our work, but differs in that they fuse all embeddings as a central embedding and then minimize distances, whereas we align each modality separately.

## 3 Problem Description

Given a language command (either text or speech) that describes an object, we want our model to retrieve the correct object from a set. This problem is an exemplar of tasks found in the area of grounded language learning in the fields of robotics and natural language processing. Intuitively, the goal is to take unconstrained natural language queries and select the appropriate object based on the complete set of sensor inputs available to the agent. We demonstrate on a domain containing four modalities, all of which refer to objects in the environment: spoken language, written text, RGB (image) inputs, and depth camera inputs.

More formally, given a spoken language command $x_s$, a textual language command $x_t$, a set of RGB images $X_r = \{x_r^{(1..n)}\}$, and a set of depth images $X_d = \{x_d^{(1..n)}\}$, the task is to retrieve the correct object by choosing

the index that has the minimum distance from either of the language commands across all modalities. Depending on which modalities are ablated, we consider up to four distances: $sr$, a vector of distances between $x_s$ and all RGB images in $X_r$; $sd$, a vector of distances between $x_s$ and all depth images in $X_d$; $tr$, a vector of distances between $x_t$ and all RGB images in $X_r$; and $td$, a vector of distances between $x_t$ and all depth images in $X_d$. In order to select the correct object, we first perform a component-wise average of the relevant modality pair distances for the available modalities. Then, we select the object which has the minimum distance, i.e., we perform an argmin on this average vector of multiple-modality distances. Depending on the missing/available sensors during test time, we might have any combination of these four distances. For example, if no written instructions are available at test time,[1] we compute the component-wise average of $sr$ and $sd$, and then select the object whose coordinate resulted in the lowest average distance. This method allows us to extend our model to support arbitrary modalities while remaining robust to test cases in which some modalities are missing or incomplete.

## 4   Approach

In keeping with previous work on the closely related problem of image retrieval, we focus on contrastive loss approaches, in which the goal is to learn an embedding of data where similar samples—in our case, samples belonging to the same class of object—are embedded 'close' to one another, while dissimilar samples are farther apart. Since contrastive learning is based on similarity, it is aligned with our goal of training a multimodal model for grounded language learning: contrastive learning enables us to teach the model to map different modalities of the same object closer to each other in the latent space. We describe a novel geometric loss function, GEOMETRIC ALIGNMENT, that simultaneously minimizes intra-class distances and maximizes inter-class distances across each pair of modalities, yielding a model that is both effective at the retrieval task defined above and robust to modality dropouts at test time. We additionally combine this GEOMETRIC ALIGNMENT loss with a classification-based (cross-entropy) loss function which results in a superior model compared to either geometric or cross-entropy losses alone; we refer to this combination as Extended Multi-Modal Alignment, or EMMA.

*Core Concepts.*   As per typical contrastive loss, we refer to three primary kinds of sample: *anchors*, data points that acts as a point of reference; and *positives* and *negatives*, which refer to a data point(s) that are similar and dissimilar to the anchor, respectively. For example, to learn the meaning of the concept "book," the anchor might be an RGB image of a book; the corresponding text description and depth image of that book are in the positive set; and the negative set includes the text description and RGB image of an apple. The methods we discuss differ both in how they choose these three sets and the objective function used.

### 4.1   Baselines

We compare both EMMA and GEOMETRIC ALIGNMENT against contrastive learning (Chen et al., 2020) and supervised contrastive learning (Khosla et al., 2020), which for conciseness we refer to as SUPCON. We consider SUPCON as the main baseline since it is a general version of multiple contrastive loss functions including triplet loss and N-pair loss (Sohn, 2016).

### 4.1.1   Contrastive Loss

We compare our model against the contrastive learning method presented by Chen et al. (2020) where they use the normalized temperature-scaled cross-entropy loss (NT-Xent). In order to implement this loss function, we use cosine similarity as suggested in Chen et al. (2020). Another possibility is to use an inner dot product (Khosla et al., 2020); if not normalized, this can lead to numerical instabilities and overflow/underflow since the dot product is not bounded, but the normalized inner dot product is similar to cosine similarity. We can treat different modalities of an instance as different augmentations of it and consider any combination of two of them as a positive pair. For each item $i$ in a batch and each positive pair

---

[1]This setting is particularly salient. While large bodies of text are frequently available at training time, a person interacting directly with a physical agent may well prefer to use only spoken instructions.

of $i$ and $j(i)$ the contrastive loss function is then formulated in eq. (1),

$$-\log \frac{\exp(sim(z_i, z_{j(i)})/\tau)}{\sum_{a \in A(i)} \exp(sim(z_i, z_a)/\tau)}, \tag{1}$$

where $A(i)$ are all negative and positive indices (except $i$), $sim$ is a similarity function, and $z = f(x)$.

### 4.1.2 Supervised Contrastive Learning

Khosla et al. (2020) extend the contrastive learning method (NT-Xent) and propose a supervised way of performing contrastive learning to treat not only augmentations of the anchor but also every item that shares the same label with the anchor as positives. This loss function is shown in eq. (2):

$$\sum_{i \in I} \frac{-1}{|P(i)|} \sum_{p \in P(i)} \log \frac{\exp(z_i \cdot z_p/\tau)}{\sum_{a \in A(i)} \exp(z_i \cdot z_a/\tau)}. \tag{2}$$

Although this loss function does not use cosine similarity, embeddings are normalized before performing the dot product, which is equivalent to cosine similarity.

The authors applied this baseline to a unimodal dataset. In this paper, we extend it to work with a multimodal dataset and show that it is slower than EMMA to learn and requires more data to converge. Since SUPCON considers all pairwise distances in each batch, with $M$ modalities and a batch of size $B$ each batch contains $BxM$ items, and SUPCON computation involves $(BM)^2$ pairwise distance terms. However, the computations of our GEOMETRIC ALIGNMENT approach are agnostic with respect to batch size which makes it scalable. As originally proposed, the SUPCON approach we choose as our baseline was applied to unimodal datasets such as ImageNet (Deng et al., 2009), CIFAR-10, and CIFAR-100 (Krizhevsky & Hinton, 2009). We both demonstrate that it is possible to use SUPCON for multimodal datasets and extend it with the addition of a geometric component, which gives benefits to performance and training speed.

### 4.2 EMMA: Extended Multimodal Alignment

Our proposed multimodal method is composed of two complementary parts. The first part is a geometric loss based on distances in the latent space, and the second is a supervised contrastive loss based on cross-entropy (SUPCON). The geometric loss is faster to learn and is more aligned with the downstream task of object retrieval, while the SUPCON loss gives the model the classification ability when the model is trained on enough data. Hence we combine them into a single model, EMMA.

### 4.2.1 Geometric Alignment Loss

We define a distance-based loss function that can be used for an arbitrary number of modalities. Our proposed method is inspired by similarity-based triplet loss, and is similar to contrastive loss (Chen et al., 2020; Khosla et al., 2020) under some settings. Triplet loss-based learning works by forcing similar concepts from different domains 'together' in some shared embedding space while forcing dissimilar concepts 'apart.' However, standard triplet loss cannot be used for more than two modalities. We note that triplet learning often require an expensive *mining* step to find harder negative samples to enable effective learning (Hoffer & Ailon, 2015; Schroff et al., 2015; Zhao et al., 2018; Zhai et al., 2018). This is problematic for throughput and scalability, as run time becomes quadratic in batch size. Our approach does not require mining to obtain good performance, alleviating these issues along with their complicated history in reproducibility and parameter tuning (Musgrave et al., 2020; Raff, 2021; 2019).

To address these issues, we use pairwise distance optimization among all modalities of an instance and a sampled negative example (rather than fixed negative examples). We modify the concept of triplet loss as follows. In training, we sample two different instances and their corresponding representations from all modalities into two sets—a positive set (referring to a specific object) and a negative set (referring to some other object), as shown in Figure 1. Unlike prior triplet loss methods, the anchor is not randomly chosen.

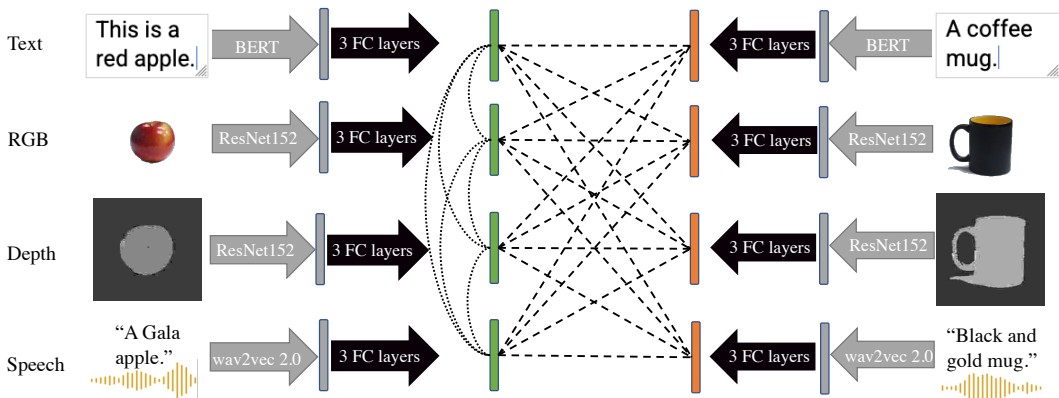

Figure 1: A high-level prototype of our approach and the distances used in the GEOMETRIC ALIGNMENT loss. Gray arrows indicate frozen pre-trained models (the parameters are fixed and not trained). The black arrows show 3 fully connected layers with a ReLU activation function (Nair & Hinton, 2010) after the first two layers. These networks are trained. Orange rectangles are negative embeddings and green rectangles are positive embeddings. Dashed lines indicate distances to be maximized while dotted lines are minimized.

Instead, every item in the positive set becomes an anchor once. We minimize the distance between that item and other items in the positive set while maximizing the distance from all items in the negative set.

The objective is to first minimize the distance between each pair of positive points from heterogeneous modalities, and second maximize the distance between each positive and negative point from all modalities. We refer to this approach as GEOMETRIC ALIGNMENT, formulated in Equation (3) and shown in Figure 1:

$$\mathcal{L} = \sum_{m_1=1}^{M} \left[ \sum_{m_2=1}^{M} \left[ -\max(dist(z_{m_1}^+, z_{m_2}^-) + \alpha, 0) \right] + \sum_{m_3=m_1+1}^{M} \left[ \max(dist(z_{m_1}^+, z_{m_3}^+), 0) \right] \right]. \tag{3}$$

In eq. (3), $M$ is the number of modalities, the superscripts $+$ and $-$ represent positive and negative objects, $\alpha$ represents an enforced margin between each positive and negative point (set to 0.4 for all modalities without tuning), and $z$ is the embedding from applying a mapping function $f$, which is a neural network on our input data. In other words, $z_m = f_m(x_m)$, where each modality $m$ has a specific model $f_m$ that is different from the models for other modalities. These models do not share their weights. Cosine similarity is the opposite of distance, and we need to reverse the logic for maximization and minimization. We use cosine similarity between pairs of embeddings. It is a good choice for high-dimensional data as it is bounded between -1 and 1. Other distance metrics, such as Euclidean distance, grow in value with respect to their dimensionality, resulting in very large distances for data points. The $\cos(\cdot)$ function is a measure of similarity, not distance.

To make the equations more readable, the generic *dist* function is replaced with the specific $\cos(\cdot)$ function. Also, we use $\text{push}(x, y) = \max(\cos(x, y) - 1 + \alpha, 0)$ for maximizing all unique pairwise distances between modalities of positive and negative instances, and $\text{pull}(x, y) = \max(1 - \cos(x, y), 0)$ for minimizing the unique pairwise distances among the modalities of positive instances. Hence, our GEOMETRIC ALIGNMENT loss function in eq. (3) can be written as $\mathcal{L} = \sum_{m_1=1}^{M} \sum_{m_2=1}^{M} \text{push}(z_{m_1}^+, z_{m_2}^-) + \sum_{m_1=1}^{M} \sum_{m_3=m_1+1}^{M} \text{pull}(z_{m_1}^+, z_{m_3}^+)$.

Our proposed GEOMETRIC ALIGNMENT loss function in eq. (3) can be rewritten as in eq. (4) by fully specifying the summations, showing how our objective function can be reduced to triplet and pairwise loss.

$$\mathcal{L} = \sum_{i=1}^{M-1} \sum_{j=i+1}^{M} \text{pull}(z_i^+, z_j^+) + \text{push}(z_i^+, z_j^-) + \text{push}(z_i^-, z_j^+) + \sum_{i=1}^{M} \text{push}(z_i^+, z_i^-). \tag{4}$$

If $M = 2$ (the number of modalities is 2) and we ignore the last two terms in the derived objective function, it results in the triplet loss method. If $M = 2$, then our objective function reduces to the quadruplet loss method (Tursun et al., 2021; Chen et al., 2017) if we multiply the first term by 2, ignore the third term, and

change the last summation to be up to $M - 1$ (which results in a single term). If $M = 1$, only the last term remains in the loss function which is exactly the pairwise distance-based loss function. This loss function can be seen as a contrastive loss, usually used in the domain of self-supervised learning (Chen et al., 2020).

Our proposed loss function has two advantages over the traditional loss expressed in eq. (1). The first is that our loss function does not need to compute the distance between all items in the batch as opposed to SUPCON (Khosla et al., 2020). Instead, we sample only two objects (positive and negative), each of which has $M$ modalities, giving $2M$ data points (or embeddings). Hence, our model can be trained using smaller batch sizes and reduces the number of negative samples we need. The second advantage is that this loss function can be used in a multimodal setting with an arbitrary number of modalities. Although our GEOMETRIC ALIGNMENT is quadratic in the number of modalities, we observe that experimentally, training time increases only by 10 minutes with each additional modality. It is noteworthy that our training procedure does not perform any stochastic dropout of modalities to obtain test-time robustness to missing modalities.

### 4.2.2 Combining Geometric Loss and Cross-Entropy based SupCon Loss

The main difference between GEOMETRIC ALIGNMENT and SUPCON is that in GEOMETRIC ALIGNMENT we focus on the geometric interpretation of similarity using cosine distances, while in SUPCON, cosine distances are used to compute a classification-based loss function similar to cross-entropy. Each method offers unique advantages. In GEOMETRIC ALIGNMENT, the advantages are an intuitive learning objective in terms of distance, interpretability of the learned embedding space, and faster convergence. The advantage of SUPCON is that with enough training data points, it provides the model with complimentary classification-based signals which can be helpful, especially with more chaotic modalities such as speech (see fig. 7). We next combine the GEOMETRIC ALIGNMENT defined in eq. (4) with SUPCON defined in eq. (2), and refer to this approach as EMMA, for extended multimodal alignment. The combination is done by taking the sum of the equations for GEOMETRIC ALIGNMENT and SUPCON (shown in eq. (5)).

$$
\mathcal{L} = \sum_{i \in I} \Bigg[ \overbrace{\overbrace{\sum_{j=1}^{M-1} \sum_{k=j+1}^{M} \mathrm{pull}(z_{i,j}^+, z_{i,k}^+)}^{\text{Align different modalities of similar objects}} \overbrace{+ \mathrm{push}(z_{i,j}^+, z_{i,k}^-) + \mathrm{push}(z_{i,j}^-, z_{i,k}^+) + \sum_{j=1}^{M} \mathrm{push}(z_{i,j}^+, z_{i,j}^-)}^{\text{Push dissimilar representations apart}}
$$

Enforcing Representational Similarity for Multi-modal Views of the Instances

$$
+ \underbrace{\sum_{m=1}^{M} \frac{-1}{|P(i,m)|} \sum_{\beta \in P(i,m)} \log \frac{\exp(z_{i,m} \cdot z_\beta / \tau)}{\sum_{\gamma \in A(i,m)} \exp(z_{i,m} \cdot z_\gamma / \tau)}}_{\text{Grounding through Contrastive Predictive Loss}} \Bigg] \tag{5}
$$

where $A(i,m)$ includes all items in the batch except for the $z_{i,m}$ itself and $P(i,m)$ includes all the modalities of all instances that have the same label as current instance excluding $z_{i,m}$ itself. In other words, the collection of embeddings indexed by $P(i,m)$ is $\{ \bigcup\limits_{r \neq m \in M} z_{i,r}, \bigcup\limits_{l \neq i \in I, y_i = y_l} \bigcup\limits_{m=1}^{M} z_{l,m} \}$.

Although both SUPCON and GEOMETRIC ALIGNMENT are trying to bring target objects together (and push dissimilar ones away), there is a trade-off in SUPCON where there is a normalized ranking across them. In contrast, in GEOMETRIC ALIGNMENT distances can be arbitrarily near or far. Our approach tries to respect the distinct modalities of each instance. While our formulation is similar to SUPCON notationally, conceptually SUPCON uses different augmentations of RGB images as positive examples (coming from the same underlying empirical distribution), while we use different modalities drawn from different distributions.

### 4.3 Network Architecture

Transformers (Vaswani et al., 2017) have become *de facto* architectures in the natural language processing community and have shown great success across different tasks. We use BERT (Devlin et al., 2019) embeddings contained in the FLAIR library (Akbik et al., 2019a;b) to featurize textual input, and

wav2vec2 (Baevski et al., 2020) to extract audio embeddings from speech. Both BERT (Devlin et al., 2019) and wav2vec2 (Baevski et al., 2020) are self-supervised language models using transformers (Vaswani et al., 2017). To process images, we use ResNet152 (He et al., 2016) for both RGB and depth images which gives us a 2048-dimensional embedding vector. Depth images are colorized before passing to ResNet152. We then use different instantiations of the same multi-layer perceptron (MLP) consisting of 3 fully connected layers with ReLU activation (Nair & Hinton, 2010) to map each of these embeddings to a shared 1024-dimensional space where we can compute the distances between all embeddings. These networks do not share weights.

## 5 Experiments and Results

In this section, we describe the experiments and provide quantitative and qualitative results by comparing our methods against supervised contrastive learning (Khosla et al., 2020) and contrastive loss (Chen et al., 2020). Moreover, we evaluate the quality of object retrieval models learned using EMMA loss function. We first describe the dataset we use and the setup of the experiments, then describe the metrics by which we evaluate performance. We demonstrate the effectiveness of our approach on a publicly available multimodal dataset called GoLD (Kébé et al., 2021), which contains RGB images, depth images, written text descriptions, speech descriptions, and transcribed speech descriptions for 207 object instances across 47 object classes (see Figure 1). There are a total of 16,500 spoken and 16,500 textual descriptions. The training set, validation set, and test set contain 7380, 4160, and 4960 items (each consisting of four modalities), respectively. Speech is converted to 16 Hz to match the wav2vec2 speech model.

### 5.1 Setup

To evaluate our model we measure different performance metrics on a retrieval task where the model selects an object from a set given a language description. Only one of the objects corresponds to the description. Similar to Khosla et al. (2020), we use a stochastic gradient descent (SGD) optimizer with momentum (Ruder, 2016) with a flexible learning rate starting at 0.05. All models are trained for 200 epochs with a batch size of 64 on a Quadro RTX 8000 GPU. Following the literature, we used a temperature of 0.1 for training the contrastive learning method (section 4.1.1), and a temperature of 0.07 for training SupCon (section 4.1.2).

We compute the distance between the given natural language description and all available sensory modalities for 5 randomly selected objects (one of which corresponds to the description, with the others from different object classes), as described in section 5.4. For example, if RGB and depth are present, we compute the distance between the language embedding and all candidate RGB and depth embeddings. We then average these distance matrices and choose the closest image embedding (average distance of RGB and/or depth from language) as the prediction. In order to use cosine *distance*, we subtract the cosine of the *angle* between two embeddings (which represents similarity) from 1.

### 5.2 Metrics

The best metric to capture the performance in such a scenario is mean reciprocal rank (MRR) for $Q$ queries. For each query, we predict the rank of all objects based on their distance from the language command, and then the inverse rank of the desired objects in all queries are averaged. We compute MRR by $\frac{1}{|Q|} \sum_{i=1}^{|Q|} \frac{1}{\text{rank}_i}$. MRR is an informative metric because it captures the idea that having the correct object as the second choice should be considered better than having it as a last choice. Because our approach is designed to be robust to missing modalities, we report MRR for different combinations of modality dropouts. Table 1 summarizes the results of all experiments. To provide a better sense of the metric, consider a model that always ranks the correct object in the second place: such a model would have an MRR of 0.5.

### 5.3 Learning from Limited Data

We consider a case where lower amounts of data are available to train the model to replicate a real-world constraint. When a personal robot is deployed, it does not have access to huge amounts of data to finetune its model and should be able to learn and adapt using the minimum amount of data possible. Figure 2 depicts

| Methods-percent | speech/depth | speech/RGB | text/depth | text/RGB | text/speech/depth | text/speech/RGB | speech/RGB/depth | text/RGB/depth | all |
|---|---|---|---|---|---|---|---|---|---|
| EMMA-25 | **70.16**±0.79 | **71.83**±0.25 | **88.44**±0.37 | **90.21**±0.49 | **87.28**±0.52 | **89.26**±0.55 | **73.07**±0.39 | **91.36**±0.41 | **90.72**±0.34 |
| SupCon-25 | 46.49±0.13 | 46.35±0.18 | 83.77±0.79 | 84.87±0.61 | 80.71±0.78 | 81.48±0.6 | 46.39±0.33 | 85.73±0.62 | 83.25±0.68 |
| Geom.-25 | 66.11±0.52 | 67.51±0.49 | 85.66±0.41 | 87.4±0.29 | 85.48±0.26 | 87.29±0.34 | 68.17±0.39 | 87.94±0.31 | 88.12±0.26 |
| SimCLR-25 | 50.99±0.52 | 51.21±0.58 | 50.37±0.43 | 50.51±0.7 | 48.59±0.21 | 48.77±0.35 | 49.13±0.56 | 48.6±0.59 | 47.75±0.17 |
| EMMA-100 | 77.63±0.29 | 78.66±0.64 | 89.87±0.5 | 91.26±0.86 | **89.66**±0.36 | 90.97±0.66 | 80.32±0.45 | 92.71±0.5 | 92.72±0.47 |
| SupCon-100 | 78.18±0.58 | **79.69**±0.54 | 89.04±0.88 | 90.56±0.74 | 88.75±0.66 | 90.5±0.69 | **81.2**±0.39 | 91.96±0.42 | 92.03±0.7 |
| Geom.-100 | 76.82±0.34 | 78.34±0.29 | 89.64±0.38 | 91.13±0.73 | 89.21±0.45 | 90.95±0.83 | 79.37±0.29 | 92.29±0.51 | 92.14±0.45 |
| SimCLR-100 | 71.74±0.73 | 73.37±0.39 | 89.72±0.54 | 90.82±0.37 | 89.13±0.61 | 90.26±0.58 | 74.96±0.44 | 91.92±0.41 | 91.72±0.53 |

Table 1: Average and standard deviation of mean reciprocal rank (MRR) over 5 runs on a held-out test set with different modalities ablated during testing. For 5 objects, MRR ranges from 20% to 100% (higher is better). A random guess would have an MRR of 0.46. The batch size is 64 and the optimizer is SGD with momentum for all experiments. Column headers describe which modalities are present at query time—for example, in column 2 'speech/depth', both text and RGB are ablated. Models were trained with either 25% or 100% of the available training data (shown in Column 1). EMMA is either better than or very close to state-of-the-art methods for most cases. With 25% of the training data used, EMMA outperforms other methods in all cases. Statistically significant results are shown in bold.

the MRR for the models trained with different amounts of training data. As expected, less training data means lower performance. However, the drop in performance is significantly higher in SupCon compared to EMMA when less data is available. With 369 training data points, SupCon does not learn and results in an MRR consistent with a random baseline, while EMMA achieves an MRR of 0.82.

While this demonstrates the empirical benefit of EMMA, it raises the question of what is contributing to this difference. We hypothesize that by decreasing the number of training examples, SupCon does not get enough examples of positive pairs in the contrastive loss. We note that the number of negative pairs in each batch is about 15 times larger than the number of positive pairs in SupCon when we have 369 training data points. However, this ratio is about 1 in EMMA. In order to test this, we reduce the batch size from 64 to 4, and the MRR score of SupCon goes from 0.47 up to 0.61.

This is an example of how carefully using minimal signals can result in a better model, similar to what we proposed in our GEOMETRIC ALIGNMENT method. This is especially important in low-data regime. SupCon, on the contrary, is useful when trained on more data since the loss function contrasts all items in the batch. When little data is available, contrasting all items in the batch confuses the model, however, with large amounts of data it leads to classification abilities since SupCon is based on cross-entropy.

The literature is not consistent about the effect of the number of negative pairs on the contrastive loss function. Awasthi et al. (2022) state that more negative examples do not necessarily hurt performance; however, they assume

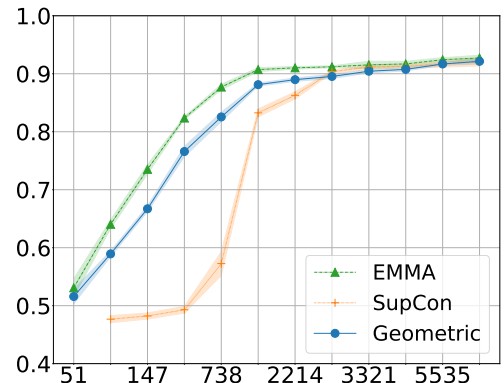

Figure 2: Mean Reciprocal Rank (*y*-axis) when all modalities are available, shown as a function of the amount of training data (*x*-axis). All averaged over 5 runs for the downstream task of object retrieval. Higher is better. Orange is SupCon, green is our proposed GEOMETRIC ALIGNMENT, and blue is our proposed EMMA loss function. We train all models for 200 epochs. EMMA outperforms other models, especially when only small amounts of training data are available.

non-overlapping latent classes. Other work (Saunshi et al., 2019; Ash et al., 2022) argues that increasing the number of negative points beyond a threshold can hurt performance, as it becomes more likely to see negative samples that share the same latent features as an anchor and positive samples. They refer to this as 'collision' and argue that it makes it harder to train the model to learn good representations. In our case, the dataset we use consists of everyday objects that share overlapping latent features (e.g., same color, same size, etc.). Hence, more negative examples when the total number of examples is low hurts learning.

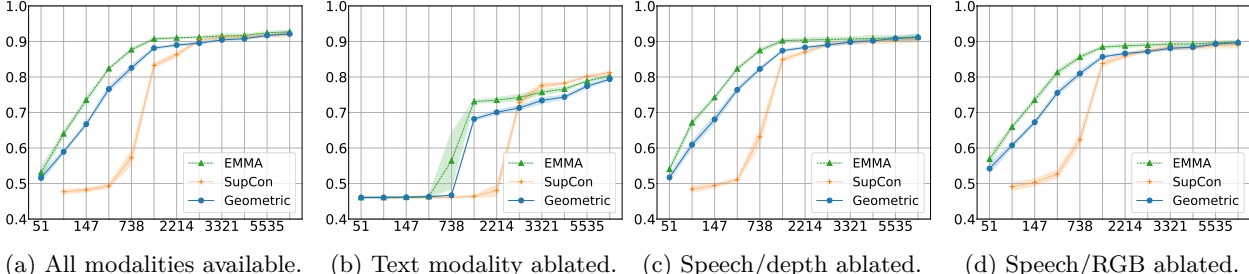

(a) All modalities available.  (b) Text modality ablated.  (c) Speech/depth ablated.  (d) Speech/RGB ablated.

Figure 3: Mean Reciprocal Rank ($y$-axis) for selected modalities of the held-out test set, shown as a function of the number of training samples ($x$-axis) and averaged over 5 runs. Higher is better. Orange is SupCon, green represents our proposed Geometric Alignment, and blue is our proposed EMMA loss function. In all cases, EMMA outperforms other models when smaller amounts of training data are available.

## 5.4 Modality Ablation

There may be different interpretations of "missing modalities" and hence different design choices. One of the advantages of our approach is that when different sensory modalities become unexpectedly unavailable during operation (test time), its performance degrades only minimally. In Figure 3 and table 1 we show that in such ablations, EMMA outperforms SupCon in low data scenarios. The loss function requires no changes beyond adjusting the value of $M$ in eq. (3) according to the number of modalities available during training. Our goal is the non-trivial downstream prediction task: determining what objects are being referred to by arbitrary language from a small number of examples. When we consider only text, RGB, and depth, written language is used as the query modality, and we compute the distance of RGB and depth modalities from it and then average them. However, when speech is incorporated as an additional modality, we compute the distance of RGB and depth from text and from speech which gives us 4 distance matrices, which we then take the average of. At test time, depending on which modalities are available to the model, we can use speech, text, or both to take the average distance of RGB and depth embeddings from the linguistic query.

There are a total of nine possible modality dropouts and corresponding distance computations. In all cases we compute the distances of available modalities (and average them if there are more than two modalities). Figure 3 shows the MRR per number of training data for four example cases in which different modalities are ablated. At 25% of the training data, SupCon struggles to learn and the converged MRR is at 0.46, while EMMA achieves an MRR of 0.73. A random model would have an average MRR score of 0.46.

When we drop the text modality (table 1), performance decreases from about 0.93 to about 0.82, showing that speech cannot completely replace text in our current effort. In Figure 7, the alignment of shared embeddings for a randomly sampled set of classes is visualized; this visualization suggests that the speech modality is not aligned as well as the text modality. This supports our hypothesis that a geometric alignment of the latent space is crucial to a good performance for object retrieval and multimodal understanding. There is a slight gap in performance when depth or RGB are dropped (Figures 3c and 3d) compared to when all modalities are available (Figures 3a and 4), showing that our model is robust when RGB or depth sensors fail.

In Figure 3b, when speech is used as the query and the text modality is ablated, the SupCon baseline works slightly better than EMMA as more training data becomes available, while EMMA still learns faster. Although the difference is negligible, the reason is that SupCon optimizes for the classification task and when enough data is available the classifier can be trained. But, Geometric Alignment is trained to align modalities properly in the latent space, which does not necessarily help the downstream task especially if one of the modalities is chaotic, and in our case, the speech modality has a high variance which makes it harder to align. However, both Geometric Alignment and EMMA work significantly better than SupCon when less training data is available. Future research will consider strategies to align more chaotic modalities.

## 5.5 Convergence Speed

Figure 4 shows the MRR per epoch of EMMA and GEO­METRIC ALIGNMENT against SUPCON (Khosla et al., 2020) and SimCLR (Chen et al., 2020) when all modalities are available. EMMA learns faster and results in better per­formance compared to both SUPCON (Khosla et al., 2020) and SimCLR (Chen et al., 2020) with all modalities avail­able at test time. We observe that not only does SimCLR learn more slowly, but that it is prone to overfitting; while this can be addressed with careful tuning, an approach that is innately robust to overfitting without tuning is preferable.

EMMA takes almost 8 epochs to converge and each epoch takes roughly 0.7 minutes, yielding 5.6 minutes until con­vergence, while SUPCON takes 36 epochs to converge and each epoch takes approximately 0.52 minutes, totaling 18.7 minutes. When we ablate modalities or use less training data, training takes less time.

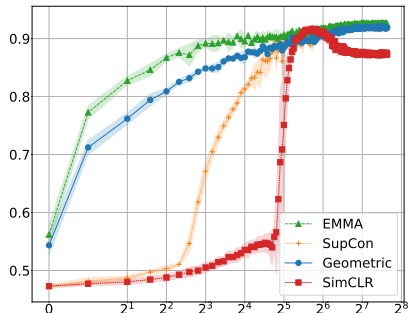

Figure 4: Mean Reciprocal Rank ($y$-axis) on the held-out test set with the ablation of se­lected modalities against training epochs ($x$-axis), averaged over 5 runs. In almost all cases, EMMA learns in fewer epochs.

## 5.6 EMMA Component Contributions

As EMMA is a combination of a geometric approach and a cross-entropy-based contrastive learning, it is worth considering what each component contributes to the algorithm's performance. One of the obvious advantages of GEOMETRIC ALIGNMENT over SUPCON is that the loss function of GEOMETRIC ALIGNMENT is designed based on distances between embeddings. This design is more aligned with the downstream task of object detection based on the distance of candidate object embeddings from the embedding of the language query. This is evident when small amounts of training data are used (fig. 2).

Figure 5 shows the impact of each of GEOMETRIC ALIGNMENT and SUPCON (represented by color) on the overall MRR of ob­ject retrieval when the weight of each component's contribu­tion to EMMA varies. The $y$-axis shows the MRR score, and a higher value means better object retrieval ability. The $x$-axis shows the percentage of dissimilar object representations being mapped outside the specified margin from each other in the latent space. A higher percentage means more of the dissimi­lar object embeddings in the latent space respect the enforced margin. The lighter colors indicate less weight for GEOMETRIC ALIGNMENT and more weight for SUPCON, and vice versa. The figure shows three positive correlations between each two com­binations of the three dimensions ($x$-axis, $y$-axis, and color).

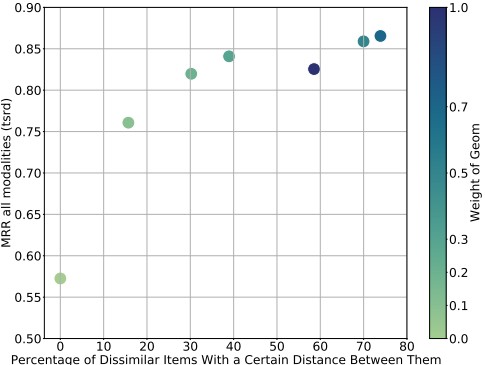

In GEOMETRIC ALIGNMENT we enforce a margin between neg­ative and anchor points to make sure that dissimilar items are not mapped within that margin from each other. In fig. 5, we see that increasing the weight of GEOMETRIC ALIGNMENT in EMMA increases both the MRR and the number of dissimi­lar data points being mapped such that the distance between them is greater than the minimum margin. Figure 5 also shows

Figure 5: The impact of GEOMETRIC ALIGNMENT and SUPCON on the MRR and the percentage of dissimilar items being mapped outside a margin from each other, as a function of their contribution (weights) in EMMA. Higher is better for both $x$ and $y$ axes.

a positive correlation between MRR and the percentage of dissimilar items mapped outside a margin from each other, which suggests that enforcing a margin between anchor and negative points in the loss function of GEOMETRIC ALIGNMENT helps the model learn better representations in the latent space.

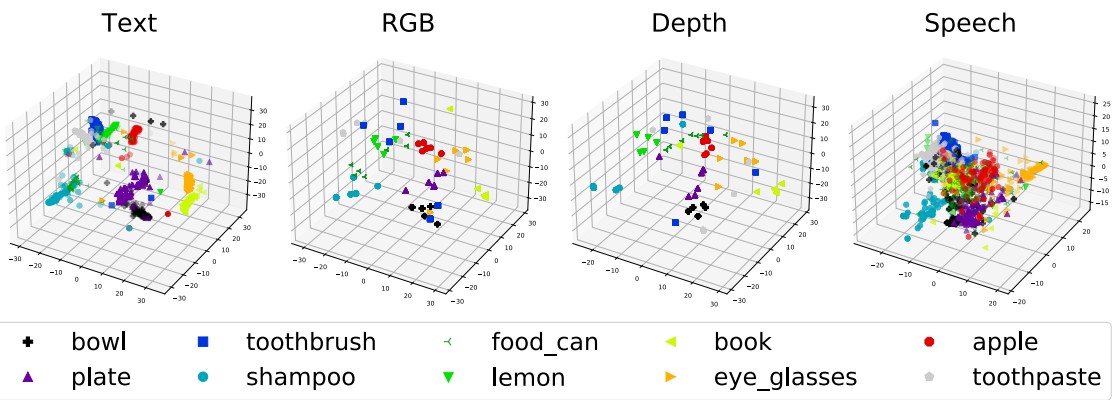

Figure 7: 3D T-SNE (Van der Maaten & Hinton, 2008) projection of test embeddings of 10 randomly selected classes of objects using EMMA. Each modality is separately projected into a three-dimensional space. Each RGB and depth image is associated with several language descriptions, leading to denser plots for text and speech. In a perfect embedding, all instances of a class would be clustered in identical areas of the embedding space across all modalities. We can see that EMMA successfully encourages all four modalities to live in a common manifold, allowing accurate retrieval even when modalities are missing.

## 5.7 Qualitative Results

An example of the need to consider multiple modalities jointly is shown in fig. 6, showing how EMMA is able to correctly select an object instance from several similarly shaped and describable objects. EMMA more accurately ranks objects and handles ambiguities with respect to retrieval queries. In a retrieval case where the query is "This is a can opener. It is light blue in color." and the candidate objects are can opener, potato, soda bottle, book, and light bulb, EMMA correctly selects the can opener as the desired object, while SUPCON selects the light bulb.

To help understand the embeddings EMMA learns, in fig. 7 we consider projections of a randomly selected subset of classes of the high-dimensional learned embeddings into a 3-dimensional space using t-SNE (Van der Maaten & Hinton, 2008).

Combined with the quantitative results, these projections demonstrate that our model is learning to map instances of the same class closer to each other regardless of their modalities. As qualitative exemplars, we note that toothbrush and toothpaste are mapped almost on top of each other in the text modality, showing semantic and syntactic similarity, but not in the visual modalities. Also, apples and lemons are mapped close to each other in all modalities.

There remains room for improvement. Specifically, the speech modality is harder to handle. Figure 7 shows that although the relative position of instances is correct in the speech space, the distinction and clustering of different objects are not as good as the other three modalities. Text is better clustered, which makes sense, as the variation in written text is much smaller than the other three modalities. Variation in speech is higher because there are a number of factors affecting speech understanding, including different accents, native language, gender,

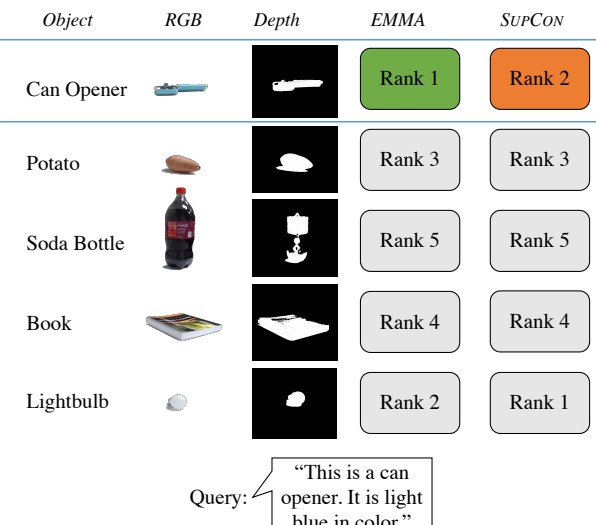

Figure 6: An example of the downstream task of object retrieval. The green rectangle shows a correct retrieval and the orange rectangle indicates a wrong one. Both models are trained on only 25% of the training dataset using the same setup. While SUPCON confuses "light blue" in the query with "light bulb" and predicts light bulb, EMMA correctly identifies the can opener as the intended object.

and age (Kébé et al., 2022). Also, it will be valuable to explore using the presented system to support object retrieval performed by an embodied (real or simulated) agent Mo et al. (2023); Bejjani et al. (2021).

# 6 Conclusion

In this work, we have demonstrated the effectiveness of our approach to learning from high-dimensional information from an arbitrary number of input modalities, both when training data is scarce, and when one or more modalities are unavailable at test time. On an object retrieval task from a testbed that contains four separate modalities, consistent with the information that might be available to a physical agent, our model EMMA achieves significantly better results when only small amounts of training data are available. This gap is even larger when the text modality is missing during the test and speech is used as the query. Moreover, EMMA is more than twice as fast as SUPCON and requires about 30% of the time that SUPCON needs to converge. The requirements for agents in different settings may differ, and training speed may not be critical in offline learning scenarios; however, since models often need to be fine-tuned for different tasks when it comes to transfer learning, the training speed is relevant. More importantly, in scenarios where language labels are provided by human interaction, learning from small amounts of data helps address a potential bottleneck in the learning process. Our proposed method is general enough to be applied to a variety of multimodal retrieval problems.

## Acknowledgments

We would like to thank the anonymous reviewers for their helpful comments, questions, and suggestions. This material is also based on research that is in part supported by the NSF under Grant Nos. 2007290, 2024878, and 2145642; the Army Research Laboratory, Grant No. W911NF2120076; and by the Air Force Research Laboratory (AFRL), DARPA, for the KAIROS program under agreement number FA8750-19-2-1003. The U.S. Government is authorized to reproduce and distribute reprints for Governmental purposes notwithstanding any copyright notation thereon. The views and conclusions contained herein are those of the authors and should not be interpreted as necessarily representing the official policies or endorsements, either express or implied, of the Air Force Research Laboratory (AFRL), DARPA, or the U.S. Government.

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
