# OpenReview forum: "Multimodal Language Learning for Object Retrieval in Low Data Regimes in the Face of Missing Modalities"
_TMLR — Accepted by TMLR_

### Review · Reviewer_5qzW · 2023-08-21

**Summary Of Contributions:**

This paper proposes an extended multimodal alignment (EMMA) to learn multimodal retrieval models. The authors propose a novel loss function that combines the supervised contrastive loss and modified triplet loss (geometric alignment), and extend its application to the realm of multimodal scenarios. The conducted experiments show that EMMA converges faster, achieves better results under constrained data conditions, and comparable or slightly improved results when more data is accessible.

**Audience:**

Yes

**Broader Impact Concerns:**

The reviewer did not see any major ethical concerns in this paper.

**Claims And Evidence:**

Yes

**Requested Changes:**

See the weaknesses above, especially justifying why training speed and limited data matter with concrete problem settings.

**Strengths And Weaknesses:**

Strengths:

1. This paper extends the contrastive loss to a multimodal setting where an arbitrary number of views and certain modalities can be unavailable.
2. Extensive experiments show that EMMA can converge faster than prior methods and learn better representations when limited data is available.
3. The paper is well written and easy to follow.

Weaknesses:

1. The proposed method shows better results when less data is available (e.g., 369 data points). Is the training data the bottleneck for agents in a real world scenario?

[Updated] The authors have addressed concerns and it will be good if they can clarify the motivation in the final revision.

2. "At 25% of the training data, SupCon struggles to learn and the converged MRR is at 0.46, while EMMA achieves an MRR of 0.73. A random model would have an average MRR score of 0.46." => Using 25% of the training data, SupCon performs as a random model?

[Updated] According to the experimental results and analysis, it seems that objectives based on contrastive learning are not suitable for low-data settings. It will be beneficial if the authors could incorporate some baselines capable of handling low-data situations and missing modalities effectively. This would enhance the soundness of the proposed model.

3. The authors claim EMMA achieves faster convergence speed (5.6 minutes) compared to SupCon (18.7 minutes). As all the models are trained offline, it is unclear to see the benefits of faster training speed, especially SupCon can achieve comparable results with a much simpler formula.
4. "Although both SupCon and the Geometric Alignment approaches are trying to bring target objects together (and push dissimilar ones away), there is a trade-off in SupCon where there is a normalized ranking across them. In contrast, in Geometric Alignment distances can be arbitrarily near or far." Can you elaborate the difference between the SupCon and triplet loss?

[Updated] Thanks for the clarification.

5. In figure 7, can you provide the 3D T-SNE of the ablations of geometric loss and SupCon loss?

---

> ### Author Response · Authors · 2023-09-15
> **Thank you for your thoughtful review and great questions**
>
> We thank the reviewer for their positive comments and their good questions. We will address them below.
>
> 1. **As to when training data is the bottleneck** in real-world scenarios, when robots are deployed in human settings, they will inevitably encounter novel environments or unfamiliar variations on known tasks, and should be able to adapt to the environment by learning from user-specific language, and objects that are present in the environment. This process will have to take advantage of the small number of objects encountered in the real world and the small amounts of data that can realistically be gathered from human interaction. We will clarify this use case in the paper's motivation.
>
> 2. The reviewer is correct that at sufficiently small amounts of data, SupCon performs essentially randomly. As an explanation, in section 5.3 we note that "When small amounts of data are available, contrasting all items in the batch confuses the model." We performed a literature review to see if this is supported by other researchers, and the answer was positive. In the paper we mention that "(Saunshi et al., 2019; Ash et al., 2022) argue that increasing the number of negative points beyond a threshold can hurt performance, as it becomes more likely to see negative samples that share the same latent features as an anchor and positive samples."
>
> 3. **There are two main benefits of faster training speed**. The first is that although our models are trained offline (so efficiency may not be the top priority), when models are deployed on robots, they may require  further training as they adapt to their environments, and faster training is crucial to having a more user-friendly agent. Moreover, other things being equal, shorter training times reduce the carbon footprint and are more environmentally friendly.
>
> 4. The differences between SupCon and triplet loss are as follows:
>     - Triplet loss requires three data points (an anchor, a positive, and a negative), however, in SupCon, all items in the batch are contrasted.
>     - SupCon can be extended to more than two modalities, but triplet loss is not well-defined for more than two modalities.
>     - SupCon implicitly models a distribution, while triplet loss is purely distance-based.
> \
> \
> For these reasons, we introduce the Geometric Alignment approach. While similar to triplet loss, the Geometric Alignment approach is different from it. The Geometric Alignment approach resolves all of these issues to have the best of both worlds, namely:
>     - Our Geometric Alignment approach can be extended to incorporate any number of modalities (as opposed to triplet loss).
>     - Contrary to SupCon, our Geometric Alignment approach does not contrast all items in the batch, and therefore, it is more memory and time efficient. Moreover, a smaller number of negative examples and their careful selection prevents confusing the model by providing contradictory signals (especially since the objects share the same latent features).
>
> 5. If space permits, we will include the 3D T-SNE of the ablations in the final version. However, unlike EMMA, SupCon does not provide a well-separated and aligned latent space when visualized with 3D T-SNE, consistent with our other experimental results.

---

### Review · Reviewer_LDTU · 2023-08-23

**Summary Of Contributions:**

**[Updated]** This paper presents EMMA, a method for learning joint representations of multiple modalities (images, text, speech) with limited data and missing modalities. EMMA integrates contrastive learning with a geometric alignment loss to learn representations that minimize distances between instances and maximize distances between classes. Experiments are conducted on the GoLD dataset, which consists of simple language-grounding classification tasks with multi-modal inputs: RGB images, depth images, text descriptions, and speech descriptions. Results indicate that EMMA substantially outperforms prior methods like contrastive-learning and supervised-contrastive-learning in low-data regimes.

**Audience:**

Yes

**Broader Impact Concerns:**

The reviewer does not see any major ethical or moral concerns regarding the method or dataset used in the paper.

**Claims And Evidence:**

Yes

**Requested Changes:**

**[Updated]** No major changes. But nice to have:
- Some experiments outside classification datasets. Perhaps with some object-retrieval task in an embodied simulator.

**Strengths And Weaknesses:**

Strengths
- **[Updated]** EMMA achieves compelling quantitative gains over prior methods in low-data regimes. These new results (in Table 1) fit much better with the narrative in the introduction.
- **[Updated]** The paper presents an ablation study investigating the contributions of individual components in EMMA. This new analysis is very helpful in understanding how EMMA works.
- **[Updated]** The introduction, methods, and results sections have greatly improved in writing. The context for low-data multimodal learning is much better motivated.
- **[Updated]** The authors have done a good job of incorporating feedback from previous reviews.
- EMMA tackles an important problem in real-world perception. Robots and other embodied agents in physical environments often suffer from sensory failures and occlusions. Learning joint representations of multiple modalities that are robust to modality dropouts could potentially improve the reliability of such systems.
- Figures 3 and 4 are particularly informative. The qualitative results in Figure 4 could be further explored in future works to study the generalization capabilities of multi-modal representations over uni-modal representations.

Weaknesses
- Despite being motivated by agents with faulty sensors, the choice of the GoLD dataset – a static classification dataset – seems odd. For embodied agents, classifying among 5 possible objects is rarely a relevant problem formulation.

---

> ### Author Response · Authors · 2023-09-15
> **Thank you for your positive feedback and constructive comments**
>
> We thank the reviewer for their positive comments. We are glad that the reviewer finds our new analysis helpful in understanding how EMMA works. We are also glad that our changes and consideration of feedback was well received.
>
> It is true that the GoLD dataset is static and does not contain noise in the form of objects that are in motion or changing rapidly. Therefore, to simulate faulty sensors we apply sensor dropouts during the test time. Regarding the use of a static dataset, in household environments, robots have to deal with objects that they consider from moment to moment. In other words, objects are not **dynamic**, and the GoLD dataset is a good representative of everyday objects that are static as a robot takes snapshots of the environment. We acknowledge that the word *static* can mean different things to different communities, and we hope we addressed what the reviewer had in mind.
>
> We agree that for embodied agents classifying among five objects might not always be the most relevant case, however, at any given time, the agent has a limited field of view.
> Moreover, our setup is designed to scale to more than five objects, and if a baseline is more confused than our model when classifying among five objects, adding more objects will decrease the performance. In low-data settings (e.g., 147 instances available for training), the baseline performs poorly (only slightly better than random), while our model is doing a decent job. However, by having a well-established and straightforward task, we can compare models against more baselines.
>
> Re: "Nice to have": We agree with the reviewer that object retrieval by a robot (embodied agents) is a potentially exciting target for this work, and this is also supported by the recent works in the literature [1,2]. We will discuss this in the paper accordingly, e.g., "In future it will be valuable to explore using the presented system to support object retrieval performed by an embodied (real or simulated) agent [1,2]."
>
> [1] Y. Mo, H. Zhang and T. Kong, "Towards Open-World Interactive Disambiguation for Robotic Grasping," 2023 IEEE International Conference on Robotics and Automation (ICRA), London, United Kingdom, 2023, pp. 8061-8067, doi: 10.1109/ICRA48891.2023.10161333.
> [2] W. Bejjani, W. C. Agboh, M. R. Dogar and M. Leonetti, "Occlusion-Aware Search for Object Retrieval in Clutter," 2021 IEEE/RSJ International Conference on Intelligent Robots and Systems (IROS), Prague, Czech Republic, 2021, pp. 4678-4685, doi: 10.1109/IROS51168.2021.9636230.

---

### Review · Reviewer_NCbe · 2023-09-02

**Summary Of Contributions:**

This paper studies the setting where there are more than two modalities (one of them a natural language query and other modalities representing physical objects), in a low data regime, and with the real-world constraint that sensors fail due to obstructions or adverse conditions. The authors propose a method called extended multimodal alignment (EMMA) that uses a combination of geometric alignment loss and supervised contrastive learning loss. Results show strong results in grounded language object retrieval based on four modalities including vision, depth sensing, text, and speech, and faster convergence.

**Audience:**

Yes

**Claims And Evidence:**

No

**Requested Changes:**

see the 3 points in suggestions above.

**Strengths And Weaknesses:**

note: I had reviewed the previous version of this submission. My main concerns from the previous version were:

1. It was not clear when the proposed approach (using both geometric and contrastive alignment objectives are useful) actually works, and there should be a deep discussion on when we need alignment, when geometric and contrastive alignment is necessary, and when we should combine both.

---> The authors have addressed this concern with through adding section 5.6 which varies the weight of the two components of EMMA (Geom and SupCon) to study their effects on performance and other qualitative measures of the representation space. The results here are quite useful and can be broadly used in the community for investigating various distance measures and supervised/unsupervised versions of contrastive learning. These experiments also show the effectiveness of their proposed methods which combine both objectives.

2. The results were not very convincing, and there was a lack of standard deviation and statistical analysis.

---> The authors have addressed this partially by focusing on the low data regime where it seems likes performance boosts are higher, and by adding multiple runs and deviations. However, there is a concern that the bolding of numbers in Table 1 is misleading, since some of the bolded numbers fall into the same standard deviation as unbolded numbers. I would suggest a statistical test and only bolding if the increase in performance is significant, for example referring to https://aclanthology.org/P18-1128.pdf.

3. There were some concerns about the computational tradeoffs since you are computing and optimizing 2 objective functions, and how that affect learning. There can also be deeper analysis of how learning should be performed - ablating training for both at the same time, or 1 then the other?

---> The authors have addressed computational concerns and added running/training time details. The authors have also added suitable ablations wrt training objectives and modalities.

For the new version, I am mostly satisfied with the changes. There are some other concerns/suggestions I have for the authors:
1. The authors motivate the problem setting of many modalities, but if I'm not wrong there is nothing specifically new about their methods that make it suitable for dealing with many modalities, they just take pairwise contrastive objectives and do it pairwise for M>2 modalities. More clarification here would be useful to make the motivation more clear. There have also been some studies on models for many modalities, both in TMLR last year, which could be good references under the related work paragraph 'To the best of our knowledge, there has been limited work on incorporating more than three modalities into a general learning approach.'
https://openreview.net/forum?id=ttzypy3kT7
https://openreview.net/forum?id=1ikK0kHjvj

2. Likewise the method isn't really tailored for missing modalities, despite the authors motivating that front. It would be good to get some clarification here too, and similarly references for recent work in dealing with robustness for missing modalities should be valid baselines if that is a main contribution of the paper. For example there are some baselines for noisy/missing modalities listed in the recent multimodal survey paper: https://arxiv.org/abs/2209.03430
Right now the main comparisons are to other CL-based objectives, but this class of methods may not be the most suitable for dealing with low-data and missing modality regimes in the first place.

3. Is there something deeper you can say about the 2 objectives either theoretically or via experiments? Is it mostly about the impact of margin loss vs contrastive loss on representation space? Or is the distinction mostly about the contrastive objective being unsupervised or supervised? If its the first one I should be able to replace supCon part in the objective with unsupervised contrastive learning and still have it work well. If its the second one then I should be able to combine CL with supCon (without geometric) and still do really well. Or if I really need to do supCon and I have access to labels why don't I just directly do supervised learning/fine-tuning after I train with unsupervised geometric objective. Some clarification here would be great.

---

> ### Author Response · Authors · 2023-09-15
> **Thank you for your positive and comprehensive feedback**
>
> We thank the reviewer for their constructive feedback and great questions, and we are happy that they are mostly satisfied with the changes. We appreciate that the reviewer took their time to let us know how we resolved the concerns they had in the previous submission, and finds our results are of use to the community.
>
> **Bold values**: We performed a statistical significance test, and all the results in the low data experiments (25% of data) are statistically significant with very small p-values. For the full data, three cases are statistically significant. We will change the bolding according to these results.
>
> Responses to specific concerns/requests for clarification:
>
> 1. **Incorporating many modalities**
>     - The novelty of our method lies in the formation of contrastive sets and how we use different modalities of the objects to perform learning. This careful selection of the contrastive pairs enables our model to take advantage of the information in the low-data regime and outperform the baselines. Although pairwise loss is a well-known method, the way it is used in the loss function can make a significant difference. For example, triplet loss is also two sets of pairwise contrastive loss, but the way the contrastive pairs are selected is important.
>     - Thank you for providing related studies. We will cite the papers you mentioned and discuss them in the related work section by stating "To the best of our knowledge, there has been limited work on incorporating more than three modalities into a general learning approach. However, recent work in the literature has given more attention to this problem [3,4]. [3] provides two new metrics to measure how similar two modalities are and how different pairs of similar modalities interact. The authors propose HighMMT, a multimodal transformer-based approach that can incorporate up to 10 modalities by fusing modalities that include unique information. [4] proposes GATO, a generalized multimodal model that can perform multiple different tasks using the same network." \
> [3] https://openreview.net/forum?id=ttzypy3kT7 \
> [4] https://openreview.net/forum?id=1ikK0kHjvj
>
> 2. **The point about missing modalities**
>     - Our results on modality ablations demonstrate applicability under this experimental regime. We apologize if our terminology was not clear. There may be different interpretations of "missing modalities" and hence different design choices. We address the case when perceptual modalities may become unexpectedly unavailable during operation. By being robust against this case, we mean that our model can still work if the modalities are not available at test time.
>     - Thank you for providing the recent multimodal survey paper; we will make sure to cite this new survey in addition to the previous survey in the final version.
>     - **Regarding the comparisons against non CL-based objectives**: We agree that other non-contrastive learning approaches might be equally suitable to address this problem. Since contrastive learning is based on similarity, it is aligned with our goal of training a multimodal model for grounded language learning: CL enables us to teach the model to map different modalities of the same object closer to each other in the latent space. Our empirical results indicate that our model is an effective approach. As mentioned in our related work section, VAEs and transformers are also promising models, and future work can examine how these approaches could handle missing modalities and low-data regimes.
> 3. **Deeper comments about the two objectives**
>     - The reviewer is correct that the margin loss is playing an important role (per section 5.6). Although we have not experimented with combining EMMA with unsupervised contrastive learning, our expectation is that we should be able to replace SupCon with unsupervised contrastive learning with decent results. However, an unsupervised baseline (SimCLR) alone is not as strong as SupCon alone (as shown in table 1 in the paper), therefore the combination of an unsupervised baseline with the geometric alignment may not be as good as the combination of SupCon and the geometric alignment.
>     - Both of your hypotheses are correct. The impact of margin loss is an important factor, but it does not mean that the supervised nature of SupCon is not playing any role. As shown in figure 5, even a small contribution from the geometric alignment can have an impact on the score; when the weight of geometric alignment is increased from 0.0 to 0.1, the MRR score goes from 0.57 to 0.76. This shows the impact of geometric alignment (margin loss) in our objective function. Geometric alignment is a contrastive-based loss function, and we combine it with SupCon (as suggested in the second part of your review).

---

### Decision · Action_Editors · 2023-10-07

**Recommendation:** Accept with minor revision

**Comment:**

There is both an audience for this work and the results are sufficient to demonstrate the claims of the paper.

Most of the remaining concerns revolve around issues of presentation -- which have been acknowledged by the authors in the response -- and can be easily addressed.  If the authors choose to expand their evaluation it will strengthen the final paper but is not necessary for acceptance.

**Audience:**

The results are of interest broadly to the multimodal research community and in specific to those working in embodied settings (e.g. with sensor noise).

**Claims And Evidence:**

The work proposes an approach for incorporating multiple modalities in a low-data regime. The evaluation is motivated by embodied settings where such signals are available but often corrupted by sensor noise.  The evaluation presented emulates such a scenario in a multiple-object choice setting and demonstrates statistically significant improvement over existing approaches.

While there are additional baselines or data regimes that could be used to further strengthen results, the reviewers are happy with recommending acceptance based on the current experiments.

---

> ### Author Response · Authors · 2023-10-26
> **Thank you!**
>
> We would like to thank the action editor and the reviewers for the constructive feedback.
> We have uploaded the final camera-ready version.